# The Regulatory Functions of σ^54^ Factor in Phytopathogenic Bacteria

**DOI:** 10.3390/ijms222312692

**Published:** 2021-11-24

**Authors:** Chao Yu, Fenghuan Yang, Dingrong Xue, Xiuna Wang, Huamin Chen

**Affiliations:** 1State Key Laboratory for Biology of Plant Diseases and Insect Pests, Institute of Plant Protection, Chinese Academy of Agricultural Sciences, Beijing 100193, China; yuchao@caas.cn (C.Y.); yangfenghuan@caas.cn (F.Y.); 2National Engineering Laboratory of Grain Storage and Logistics, Academy of National Food and Strategic Reserves Administration, No. 11 Baiwanzhuang Street, Xicheng District, Beijing 100037, China; xdr@ags.ac.cn; 3Key Laboratory of Pathogenic Fungi and Mycotoxins of Fujian Province, Key Laboratory of Biopesticide and Chemical Biology of Education Ministry, School of Life Sciences, Fujian Agriculture and Forestry University, Fuzhou 350002, China; xiuna0304@163.com

**Keywords:** σ^54^ factor, enhancer-binding proteins, motility, T3SS, virulence

## Abstract

σ^54^ factor (RpoN), a type of transcriptional regulatory factor, is widely found in pathogenic bacteria. It binds to core RNA polymerase (RNAP) and regulates the transcription of many functional genes in an enhancer-binding protein (EBP)-dependent manner. σ^54^ has two conserved functional domains: the activator-interacting domain located at the N-terminal and the DNA-binding domain located at the C-terminal. RpoN directly binds to the highly conserved sequence, GGN_10_GC, at the −24/−12 position relative to the transcription start site of target genes. In general, bacteria contain one or two RpoNs but multiple EBPs. A single RpoN can bind to different EBPs in order to regulate various biological functions. Thus, the overlapping and unique regulatory pathways of two RpoNs and multiple EBP-dependent regulatory pathways form a complex regulatory network in bacteria. However, the regulatory role of RpoN and EBPs is still poorly understood in phytopathogenic bacteria, which cause economically important crop diseases and pose a serious threat to world food security. In this review, we summarize the current knowledge on the regulatory function of RpoN, including swimming motility, flagella synthesis, bacterial growth, type IV pilus (T4Ps), twitching motility, type III secretion system (T3SS), and virulence-associated phenotypes in phytopathogenic bacteria. These findings and knowledge prove the key regulatory role of RpoN in bacterial growth and pathogenesis, as well as lay the groundwork for further elucidation of the complex regulatory network of RpoN in bacteria.

## 1. Introduction

Transcription factors play a crucial role in microbial growth and response to environmental changes by regulating the expression of target genes. In bacteria, sigma (σ) factors are the most widely occurring transcription factors. They reversibly bind RNA polymerase (RNAP) to regulate the transcription of a myriad of functional genes. To initiate the RNA synthesis, σ factors guide RNAP holoenzyme to specific promoters, melt double-stranded promoter DNA strands, and stabilize them as a single-stranded open complex [1,2,3]. According to their structure and evolution, σ factors are categorized into (a) the σ^70^ family and (b) the σ^54^ family (also called RpoN) [4]. In general, σ^70^ factors regulate the transcription of target genes by recognizing the −35/−10 promoter site (upstream from the transcription start site), while σ^54^ factors regulate the transcription of target genes by recognizing the highly conserved sequence GGN_10_GC at the −24/−12 promoter site [5,6]. 

σ^54^ factors act as a multifunctional regulator of many important biological processes. In *Pseudomonas aeruginosa*, σ^54^ regulates global gene expression (680 genes) and a wide range of biological processes, such as flagella synthesis, motility, metabolism, antibiotic resistance, and virulence [7,8,9,10]. It affects the expression of nitrogen-regulated genes (*ntrBC*, *glnA*, *glnK*-*amtB*, *nirBD*, *nasA*, *nasST*, and *nosRZDFYL*) and flagellum-related genes (*fleSR*, *fliEFGHIJ*, *flhA*, *flhF*, *fleN*, *flgA*, *fliLMNOPQR*, and *flhB*) at the transcription level. Furthermore, σ^54^ also regulates the susceptibility to tobramycin, quinolones, and carbapenems [11,12,13,14]. In *P. putida*, it also affects the utilization of nitrate, urea, and uncharged amino acids as nitrogen sources, as well as of lysine, C_4_-dicarboxylates, and alpha-ketoglutarate as carbon sources [15,16]. In addition, σ^54^ factor controls bacterial growth [17,18,19], nitrogen and carbohydrate assimilation [20,21], swimming and twitching motility [22], biofilm formation [23,24], exopolysaccharide (EPS) production [23], quorum sensing [25,26], type VI secretion systems (T6SS) [27,28], virulence [29,30], environmental adaptation [31,32], and antibiotic resistance [33,34] in other bacteria. 

Unlike σ^70^ factors, the regulatory function of σ^54^ is dependent on transcriptional activators, i.e., enhancer-binding proteins (EBPs) [35]. EBPs generally contain three domains: (a) an N-terminal signal-sensing domain, whose primary function is to perceive signals and regulate the activity of transcription activators; (b) a central AAA^+^ domain, whose primary function is to interact with the σ^54^ factor and hydrolyze ATP to release energy; (c) a C-terminal DNA-binding domain, whose primary function is to bind to gene promoters [4,36,37,38]. The central AAA^+^ domain is the most conserved domain, and it is present in all EBPs. Thus, candidate EBPs can be identified using central AAA^+^ conserved domains, and their regulatory roles in bacteria can be further analyzed.

In general, bacteria only contain one or two σ^54^ factors, but have multiple EBPs. For instance, *P. aeruginosa* contains one σ^54^ factor and a group of EBPs. Different stress-related signals regulate these EBPs through their N-terminal regulatory domains [39]. Two EBPs, DdaR and MifR, interact with the σ^54^ factor to induce the transcription of dimethylarginine dimethylaminohydrolase and PA5530 genes by sensing aromatic amino acids and extracellular C_5_-dicarboxylates, respectively [40,41]. In addition, the σ^54^ factor regulates the glyoxylate pathway, ethanolamine catabolism, (*R*)-3-hydroxybutyrate, the glycine cleavage system, and pyocyanin biosynthesis via EatR, HbcR, and GcsR, respectively [42,43,44,45]. Other EBPs, including PhhR, FleQ, AlgB, FhpR, CbrB, NtrC, DctD, FleR, RtcR, PilR, SfnR1, AauR, and Sfa3, and their regulatory roles in metabolism, motility, and virulence have also been studied in *P. aeruginosa* [13,15,46,47,48,49,50,51]. These results indicated that the σ^54^ factor forms a complex regulatory network with these EBPs, and whether these EBPs have overlapping regulatory roles needs further study. Moreover, numerous EBPs and their roles have been identified in other human and animal pathogenic bacteria [52,53]. However, the regulatory roles of the σ^54^ factor and EBPs in phytopathogenic bacteria remain largely unexplored.

Phytopathogenic bacteria, such as fungi and viruses, cause economically important plant diseases and pose a serious threat to world food security. *P. syringae*, *Ralstonia solanacearum*, *Xanthomonas species*, *Erwinia amylovora*, and *Dickeya dadantii* are the most important phytopathogenic bacteria [54]. *P. syringae* causes important crop diseases, and it is a well-known model organism for plant–pathogen interaction-related study. *R. solanacearum* is probably the most destructive pathogen worldwide, and it has a very broad host range that can infect 200 plant species belonging to over 50 plant families. *X. oryzae* pv. *oryzae* (*Xoo*) and *oryzicola* (*Xoc*) are the most important bacterial pathogens of rice, resulting in a 20–50% loss of crop yield. *X. campestris* pv. *campestris* (*Xcc*) is the causative agent of black rot of crucifers and affects cultivated brassicas worldwide. *E. amylovora* causes fire blight disease of apple, pear, quince, blackberry, and raspberry, and it threatens the safe production of major fruits. *D. dadantii* causes disease mainly in tropical and subtropical environments and has a wide host range, including *Saintpaulia* and potato. In this review, we summarized the recent research on the σ^54^ factor and its regulatory functions in these phytopathogenic bacteria to enhance the current understanding of the regulatory mechanism of phytopathogenic bacteria’s motility, growth, and virulence.

## 2. The σ^54^ Factor-Mediated Transcriptional Regulation

The σ^54^ factor (RpoN) has two conserved functional domains: an N-terminal activator-interacting domain (AID) and a C-terminal DNA-binding domain (DBD). AID performs two distinct functions: (i) inhibition of polymerase isomerization and initiation in the absence of activation; (ii) interaction with EBPs and stimulation of initiation in response to activation after binding to EBPs [55,56]. Unlike σ^70^, RpoN is dependent on EBPs to regulate gene transcription. Firstly, the transcription process involves the binding of the RpoN-RNAP holoenzyme to a target gene promoter at −24/−12 base pairs relative to the start codon, followed by a closed complex formation. The DNA in this complex remains in an inactive, closed state. Secondly, EBP binds to the target gene promoter sequence at nearly −150 base pairs relative to the RpoN binding sites. Thirdly, the RpoN interacts directly with the EBP AAA^+^ domain and hydrolyzes ATP to release energy. Lastly, the EBPs assemble into a catalytically competent oligomer upon getting stimulated by a cellular signal and remodel the RpoN-RNAP promoter complex using ATP. ATP hydrolysis opens the RpoN-RNAP–DNA complex and initiates RNA synthesis [6,37].

## 3. The RpoN and EBPs in Major Phytopathogenic Bacteria

The majority of phytopathogenic bacteria contain either one or two RpoN factors (Table 1). Two copies of *rpoN*, namely, *rpoN1* and *rpoN2*, were identified in the *Xanthomonas* genome. *rpoN1* and *rpoN2* are primarily located in a phosphotransferase system and a large flagellar gene cluster, respectively [57,58,59,60]. The nucleotide sequences of *rpoN1* and *rpoN2* are not identical, and their protein sequences share 39% identity [61]. It is well known that RpoN2 interacts with FleQ, an important EBP, and regulates flagellar-associated gene transcription, which controls flagellum synthesis and swimming motility [57,61]. PilRX, an EBP type, is located in the T4P gene cluster and forms a two-component system with PilSX [62]. In *Xoo*, RpoN2 positively regulates twitching motility and virulence in a PilRX-dependent manner. The other four candidate EBPs that contain the AAA^+^ domain were identified in *Xoo* (Figure 1). PrpR is a propionate catabolism operon regulatory protein, and it can interact with both RpoN1 and RpoN2. Deletion of *prpR* decreased the expression of *prpBC* and reduced *Xoo* growth and virulence (data unpublished). PXO_03020, PXO_03564, and PXO_03965 are NtrC family proteins, and their functions have not been studied in *Xoo* so far. The RpoN2 regulatory effects on virulence and growth have also been reported in other *Xanthomonas* species [58], but the type of EBP involved in these regulatory processes remains elusive. In the *R. solanacearum* genome, *rpoN1* and *rpoN2* are present in the chromosome and megaplasmid, respectively. Interestingly, RpoN1 but not RpoN2 is required for T4P gene expression, twitching motility, and virulence in bacteria [63,64]. A single RpoN has been found in *P. syringae*, which depends on two T3SS regulators, HrpS and HrpR, to regulate HrpL-dependent T3SS gene expression and control virulence [65,66]. Unlike in *P. syringae*, HrpS activates the *hrpL* transcription by binding to RpoN, but HrpR has not been found in *E. amylovora* and *D. dadantii* [67,68,69].

In the RpoN phylogenetic tree, the major phytopathogenic bacteria were divided into two groups. Group I included *Xanthomonas* spp., and group II included *R. solanacearum*, *P. syringae*, *E. amylovora*, and *D. dadantii* (Figure 2). Group I was further subdivided into two subgroups according to RpoN1 and RpoN2. Similarly, group II was also subdivided into two subgroups: (a) *R. solanacearum* with two copies of RpoN, and (b) *P. syringae*, *E. amylovora*, and *D. dadantii* with a single RpoN. The different classifications in the phylogenetic tree indicated that single and multiple RpoN factors might have different regulatory models in phytopathogenic bacteria.

## 4. RpoN Regulates Bacterial Flagella Synthesis and Motility

Flagella are sophisticated organelles found in many bacteria where they perform functions related to motility, signal detection, biofilm formation, colonization, and attachment to host tissues. Flagellar assembly is a highly organized process that requires the temporal expression of dozens of genes, which are regulated hierarchically. In *Xoo*, a flagellar gene cluster containing over 60 contiguous genes, was identified. These genes encode functionally diverse proteins, such as structural component proteins, protein export apparatus, regulatory factors, and proteins involved in glycosylation and chemotaxis [61]. One of the σ^54^ factor genes, *rpoN2*, is located in the central region of this gene cluster. In addition, *rpoN2* was transcribed in an operon with *fleQ*, which is located downstream of *rpoN2* and encodes an EBP [61]. Further study revealed that the flagellar gene cluster is regulated under a three-tiered hierarchy by RpoN2/FleQ and σ^28^ factor FliA (Figure 3). RpoN/FleQ, as the master regulators, control the expression of σ^28^ factor FliA and flagellar structure component protein. Furthermore, FliA controls the expression of flagellin protein FliC, flagellar cap protein FliD, flagellar chaperone proteins FliS and FliTX, and anti-σ^28^ factor FlgM [61]. Moreover, RpoN2/FleQ regulates the transcription of flagellin glycosylation-related genes (*gigx1*–*gigx10*), chemotaxis-associated genes (*cheYZA*), and c-di-GMP-related genes (*PXO 06199*, *PXO 06201*, and *PXO 06202*) [59,77]. Deletions in *rpoN2*, *fleQ*, *fliC,* and *fliA* resulted in the loss of flagella and swimming motility in *Xoo*. Interestingly, the *fleQ* mutant did not show much difference in its ability to infect rice leaves compared to the wild type. In contrast, the *rpoN2* mutant caused much less severe disease symptoms and shorter lesions [61]. This result indicated that RpoN2 might regulate *Xoo* virulence in a manner independent of flagellar motility.

Another σ^54^ factor, RpoN1, was also identified in *Xoo*. Transcriptome analysis showed that RpoN1 and RpoN2 regulate more than 30 genes in flagellar regulon [59]. Interestingly, the *rpoN2* expression level was decreased in the *rpoN1* mutant; moreover, abnormal flagellum and decreased swimming motility were also reported in the *rpoN1* mutant [59]. As per a previous study, RpoN1 indirectly regulates the RpoN2 transcription in *Xoo* [59]. These results indicate that RpoN1 and RpoN2 have overlapping regulatory roles in bacterial flagellum synthesis and swimming motility in *Xoo*. 

In *X. campestris* and *X. citri,* RpoN1 and RpoN2 are also involved in flagellum synthesis and motility. RpoN2 positively regulates the transcription levels of flagellar synthesis-related genes (*filDCES*, *flhAB*) and chemotaxis-related genes (*cheABDRWY*, *motAB*). In *X. campestris*, *rpoN2* mutant lacked the typical single polar flagellum and swimming motility [58]. However, *rpoN1* mutants showed flagellar morphology and swimming motility identical to the wild-type strain, indicating that RpoN2, but not RpoN1, is required for flagellum synthesis and motility in *X. campestris* [58]. Interestingly, the transcription levels of eight flagellar biosynthesis genes (*flhF*, *flhB*, *fliQ*, *fliL*, *fliE*, *fliD*, *flgG*, and *flgB*) and bacterial swimming motility decreased in Δ*rpoN2*, but increased in Δ*rpoN1*, suggesting that RpoN2 positively, but RpoN1 negatively regulates the flagellum synthesis and motility in *X. citri* [60].

## 5. RpoN Is Required for Nutritional Metabolism and Growth

In *P. syringae*, RpoN positively regulates the transcription of two ncRNAs, *crcZ* and *crcX*, thereby regulating the utilization of multiple carbon and nitrogen sources and influencing bacterial growth [74,78]. This transcriptional regulation is activated by CbrB, which is an EBP of RpoN and belongs to the NtrC family of response regulators [79,80,81]. CrbB binds to CrbA and forms a two-component system [79]: a conserved signal transduction system that regulates the cellular carbon and nitrogen balance and plays a central role in carbon catabolite repression in *Pseudomonas* species [82]. CbrA contains a domain similar to the solute/sodium symporter family proteins, and it is typically found in bacterial sensor kinases. CbrAB directly activates the transcription of sRNAs (*crcZ* and *crcY*) from RpoN-dependent promoters, which antagonize the repressing activity of Hfq-Crc, the key regulator of the carbon catabolite repression (CCR) process, positively regulating carbon metabolism [83,84]. Furthermore, CbrAB controls the expression of alginate biosynthetic genes and *rsmA*. It is also required to accumulate the sigma factor RpoS and core metabolites of aromatic and sugar metabolism [85,86]. Importantly, CbrAB and the master nitrogen regulator NtrBC directly control C/N homeostasis by regulating the transcription of histidine utilization genes (*hut*) [87]. When histidine is the sole source of N, the CbrAB-mediated promoter activity is weak, and NtrBC plays the dominant role in activating *hut* transcription. In succinate-depleted media, CbrAB activates *hut* transcription while derepressing the translation of *hut* mRNA mediated by the Crc/Hfq complex, which is sequestrated by the CbrAB-activated ncRNAs (CrcY and CrcZ). Interestingly, deletions of *cbrA* and *cbrB* impaired swimming and swarming motility, decreased T3SS-associated genes expression, and enhanced the sensitivity to cold [86,88]. This indicated that the RpoN/CbrAB–CrcYZ–Crc/Hfq regulatory cascade system controls more important phenotypes beyond carbon and nitrogen assimilation. 

In *Xoo*, transcriptome analysis revealed that both RpoN1 and RpoN2 regulate multiple genes involved in nitrogen and carbon metabolism. Deletion of *rpoN1* significantly reduced bacterial growth in rich media M210. In *rpoN1* and *rpoN2* double mutants, severe inhibition of bacterial growth was observed compared to the *rpoN1* mutant. Interestingly, the *rpoN2* mutant growth was identical to the wild-type strain in rich medium M210 but decreased in plant-mimicking medium XOM2 [59,61]. These results suggested that both RpoN1 and RpoN2 are required for *Xoo* growth, and RpoN1 might complement the effect of RpoN2 on growth in rich media. PrpR, an EBP, was identified in *Xoo*, which is located at the propionate catabolism operon and directly interacts with RpoN1 and RpoN2. The *prpR* deletion showed decreased *Xoo* growth in M210 media. In addition, RpoN1, RpoN2, and PrpR directly regulate numerous genes involved in the citric acid cycle (TCA cycle), such as *prpB* and *prpC*, which encode methylisocitrate lyase and 2-methylcitrate synthase, respectively (data unpublished). The TCA cycle is a well-studied and important central pathway that connects almost all the individual metabolic pathways [89]. Therefore, RpoN1 and RpoN2 activated by PrpR may affect bacterial growth by regulating the TCA cycle in *Xoo*. However, unlike in *Xoo*, RpoN1 but not RpoN2 is essential for nitrogen assimilation and growth in *R. solanacearum* [63], and the EBP involved in the RpoN1-dependent regulation of the *R. solanacearum* remains unexplored.

## 6. RpoN Regulates Virulence-Associated Phenotypes

### 6.1. T3SS

T3SS is an essential virulence mechanism in bacteria; it has highly conserved structural components and participates in virulence by injecting the effector proteins into the cytosol of host cells [90,91,92,93,94]. The alternative sigma factor, HrpL, is the primary transcription factor that controls the expression of T3SS-associated genes [95]. In *P. syringae*, the *hrpL* expression requires HrpR and HrpS. It forms a heterodimer on the hrpL promoter and interacts with the RpoN-RNA polymerase holoenzyme to activate *hrpL* transcription [65,66]. Unlike most EBPs, HrpR and HrpS contain the conserved AAA^+^ domain and C-terminal DNA-binding domain but lack the N-terminal signal-sensing domain that functions in phosphorylation-dependent modulation of response regulator activity [72,96]. Previous studies have shown that HrpR and HrpS interact with RpoN via the conserved motifs of GAFTGA and GAYTGA, respectively [66]. In addition, constitutive expression of *hrpL* in the individual *rpoN*, *hrpR,* and *hrpS* mutants restored the transcription of *hrp* genes to wild-type levels. Therefore, the RpoN-dependent cascade regulation of T3SS has proven that RpoN activates *hrpL* through HrpR and HrpS interaction, regulating *hrp* gene transcription in *P. syringae*. Interestingly, *rpoN* mutant cannot produce the phytotoxin coronatine, infect the host plant, or cause HR in the nonhost plant. Additionally, the constitutive expression of *hrpL* in *rpoN* mutant restored the HR to nonhost plants but did not restore coronatine production and growth [73,97,98], indicating that the RpoN has both HrpL-dependent and -independent regulatory pathways in *P. syringae*.

Unlike in *P. syringae*, HrpS but not HrpR interacts with RpoN and activates *hrpL*, thus regulating the transcription of T3SS-associated genes in *E. amylovora* and *D. dadantii* [67,68,69]. Sequence analysis revealed that HrpS and RpoN contact the promoter sequence of *hrpL* at the −138/−125 (TGCAA-N_4_-TTGCA) and −24/−12 (GG-N_10_-TGC) regions, respectively [67]. Furthermore, a novel ribosome-associated protein, YhbH, was identified. YhbH mediates HrpL-dependent T3SS regulation by modulating RpoN in *E. amylovora*. Individual deletions of *rpoN*, *hrpS*, *hrpL,* or *yhbH* significantly decreased the transcription of T3SS genes, such as *hrpL*, *dspE*, *hrpN,* and *hrpA*, and mutants failed to elicit hypersensitive response (HR) in tobacco. On the other hand, overexpression of *hrpL* by an inducible promoter rescued the T3SS gene expression and HR-eliciting ability in these mutants [76]. 

In addition, integration host factor (IHF), a nucleoid-associated protein, is often required to enhance the interaction between RpoN and EBP and for virulence by positively regulating the expressions of *hrpL* and T3SS genes in *E. amylovora* [75]. HrpX/Y, a two-component system, is also involved in the RpoN-dependent regulation of T3SS by activating the *hrpL* expression in *D. dadantii* [69]. In our previous study, we observed that RpoN2/FleQ positively regulates flagellin glycosylation, affecting the transcription of the T3SS genes in *Xoo* [77]. *fliTX,* a hypothetical protein gene, is located upstream to *rpoN2*, and regulated by RpoN2/FleQ and FliA in *Xoo*. *fliTX* deletion downregulated T3SS genes and attenuated induction of HR in tobacco [70]. These findings indicated that RpoN2 is dependent on FleQ to regulate T3SS genes in *Xoo* positively. However, either *rpoN1* or *rpoN2* mutants induced a hypersensitive response in tobacco, which indicates that the σ^54^ factor is not required for the functionality of the T3SS in *R. solanacearum* [63]. These studies revealed diversified regulatory effects of RpoN on T3SS in phytopathogenic bacteria.

### 6.2. T4P

The T4P, a special class of bacterial surface filament, plays a crucial role in surface adhesion, motility, biofilm formation, and virulence in bacteria [99,100,101]. More than 20 genes (named *pilAX* to *pilZX*) encoding T4P structural components and putative regulators were revealed in *Xoo*. PilRX, an EBP located in the T4P gene cluster, directly interacts with RpoN2 and regulates the T4P gene transcription, including the major pilin gene, *pilAX*, and the inner membrane platform protein gene, *pilCX*. Individual deletions of *rpoN2*, *pilRX*, *pilAX*, and *pilCX* resulted in significantly reduced twitching motility, biofilm formation, and virulence [62]. These findings suggest that the RpoN2/PilRX regulatory system controls bacterial motility and virulence by regulating T4P gene transcription in *Xoo*. The RpoN1 interaction with PilRX and its involvement in the regulation of T4P genes in *Xoo* remain unknown. 

Unlike in *Xoo*, RpoN1 but not RpoN2 regulates the T4P synthesis and twitching motility in *R. solanacearum* [63]. The regulatory effect of RpoN1 on T4P genes depends on PehR, which is one of the EBPs and forms a two-component system with PehS. *pehR* deletion showed reduced bacterial swimming motility; however, the motility of the *rpoN1* mutant was identical to that of the wild-type strain. Multiple studies have shown that PehSR regulates the *fliC* expression by regulating FlhDC, the primary regulator of flagellum synthesis, regulating swimming motility [102,103]. In addition, the regulator PhcA is involved in the swimming and twitching motility by negatively regulating the *pehR* expression [63,104]. These results indicated that both RpoN1 and PehR regulate T4P gene expression and twitching motility. Furthermore, PehR plays specific roles in controlling swimming motility in an RpoN1-independent manner in *R. solanacearum*.

### 6.3. Biofilm

Biofilm is an important virulence-associated factor that promotes bacterial aggregation and surface attachment and protects bacteria from environmental stress, dehydration, and host immune responses [105,106,107,108]. The RpoN regulation in biofilm formation has been studied in *Vibrio* spp. As per the outcomes of this study, RpoN positively regulates biofilm formation in *V. cholera* [109], *V. parahaemolyticus* [110], *V. anguillarum* [23], and *V. fischeri* ESR1 [111], but negatively regulates biofilm formation in *V. fischeri* ES114 [112]. Interestingly, NtrC, one of the EBPs in *V. cholera*, inhibits biofilm formation by negatively regulating the expression of core regulator genes (*vpsR*, *vpsT*, and *hapR*) [113]. Furthermore, RpoN was shown to be unnecessary for the biofilm formation but essential for biofilm detachment in *V. alginolyticus* [114]. The positive regulation of RpoN in biofilm formation has been well studied in *Lysobacter enzymogenes* [115], *Labrenzia aggregata* [33], and *P. fluorescens* [116], but not in phytopathogenic bacteria. In our previous study, RpoN2 and PilRX promoted biofilm formation by regulating the T4P gene expression, thereby affecting virulence in *Xoo* [62]. Additionally, T4P is necessary for biofilm formation in other *Xanthomonas* spp., as demonstrated in previous studies [117,118]. These results indicated that the RpoN-dependent biofilm regulation pathway is a vital virulence regulation pathway in phytopathogenic bacteria.

### 6.4. EPS

EPSs are cell-associated or secreted outside the cell. They contain organic macromolecules, such as polysaccharides, proteins, and phospholipids in addition to some non-polymeric molecules [119,120]. They are microbial biopolymers produced under stress in harsh environments and nutrition-deprived conditions [121]. Therefore, EPS production is one of the strategies of bacteria to fight against biotic and abiotic stresses. Additionally, bacterial EPSs play essential roles in host–pathogen interactions, as well as biofilms [122]. In *X. citri*, VemR acts as a RpoN2 cognate activator [71], located in an operon flanked by *fleQ* and *rpoN2.* It encodes an atypical response regulator that contains only a receiver domain [123]. Deletion of the *vemR* gene resulted in a reduction in virulence and EPS production [71]. Moreover, VemR positively regulates flagellar biosynthesis by controlling the transcription of the rod gene *flgG* [71], but RpoN2- and VemR-mediated EPS production remains largely unknown.

## 7. Conclusions and Future Perspectives

RpoN is an important and conserved regulatory factor in a majority of phytopathogenic bacteria. Unlike other σ factors, RpoN regulates the transcription of numerous functional genes in an EBP-dependent manner. On the basis of the number of σ^54^ factors, we schematized the regulatory mechanism of RpoN in *P. syringae* and *Xanthomonas* species, which have single and double σ^54^ factors, respectively (Figure 4). In *P. syringae*, a single RpoN along with HrpS and HrpR activates HrpL-dependent transcription of T3SS, subsequently regulating bacterial virulence. RpoN also regulates *crcZX* by binding to CbrB, regulating nutritional metabolism and bacterial growth. In *Xanthomonas*, RpoN1 and RpoN2 have both unique and overlapping regulatory roles. RpoN2 regulates flagellum and T4P synthesis by interacting with FleQ and PilRX, respectively. RpoN2 also regulates the expression of *fliTX* and *vemR* to positively control bacterial virulence. Interestingly, both RpoN1 and RpoN2 interact with PrpR and control *prpBC* expression, thereby modulating the TCA cycle and bacterial growth. Moreover, RpoN1 indirectly regulates the transcription of RpoN2 with unknown EBP. 

In the past few years, our understanding of σ^54^-dependent transcription has significantly progressed owing to the structural analysis of the σ^54^-RNAP complex and the application of transcriptome sequencing technology [5,124,125]. Concurrently, more and more σ^54^-dependent EBPs and target genes have been identified using the bioinformatic method by analyzing the conserved AAA^+^ domain and special binding sites GGN_10_GC on the promoter sequence, respectively [126,127,128]. However, whether these EPSs have redundant regulatory functions and how they competitively interact with σ^54^ factors remain unknown. Furthermore, the upstream signals received by EBPs to activate σ^54^-dependent regulatory pathways remain elusive. Our previous study identified the overlapping regulatory roles of two σ^54^ factors in motility, virulence and growth, and identified three EBPs in *Xoo*. However, it is not yet known how the two σ^54^ factors work together to regulate these pathways and which EBPs interact with σ^54^ factors in these overlapping regulatory pathways. Therefore, to dissect the complex regulatory network of σ^54^ in phytopathogenic bacteria, the following research-gaps should be addressed: (1) identification of candidate EBPs and characterization of their functions, (2) characterization of the interaction of EBPs with σ^54^ factors, (3) clarification of the redundant or unique regulatory functions of EBPs, (4) identification of the upstream signals of σ^54^ factors, and (5) identification of the conserved and specific regulatory pathways of σ^54^ factors in different phytopathogenic bacteria.

## Figures and Tables

**Figure 1 ijms-22-12692-f001:**
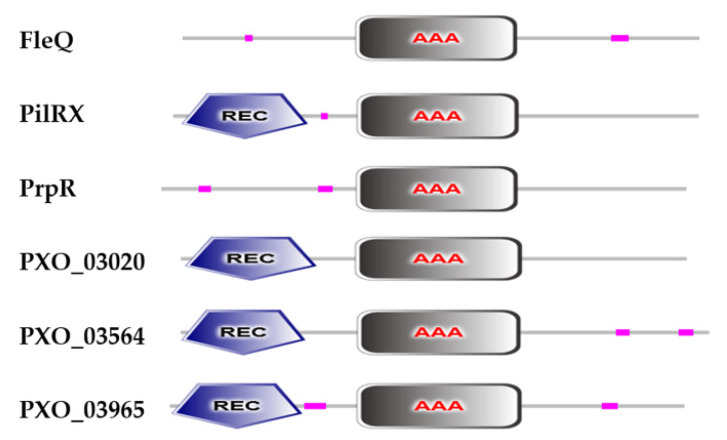
Six proteins containing an AAA^+^ domain in *Xoo*. FleQ, PilRX and PrpR are three identified EBPs of RpoN. FleQ and PilRX interact with RpoN2 to regulate the synthesis of flagella and pili, thereby regulating swimming and twitching motility, respectively. PrpR combines with RpoN1 and RpoN2 to control *Xoo* growth. PXO_03020, PXO_03564, and PXO_03965 are NtrC family proteins, and their regulatory functions have not been studied in *Xoo*. REC, response regulator receiver domain; AAA, ATPase σ^54^-interaction domain. The sequences of these proteins were downloaded from the NCBI website (https://www.ncbi.nlm.nih.gov/ (accessed on 5 May 2021)), and conserved domains were analyzed by SMART (http://smart.embl.de/smart/set_mode.cgi?NORMAL=1 (accessed on 5 May 2021)).

**Figure 2 ijms-22-12692-f002:**
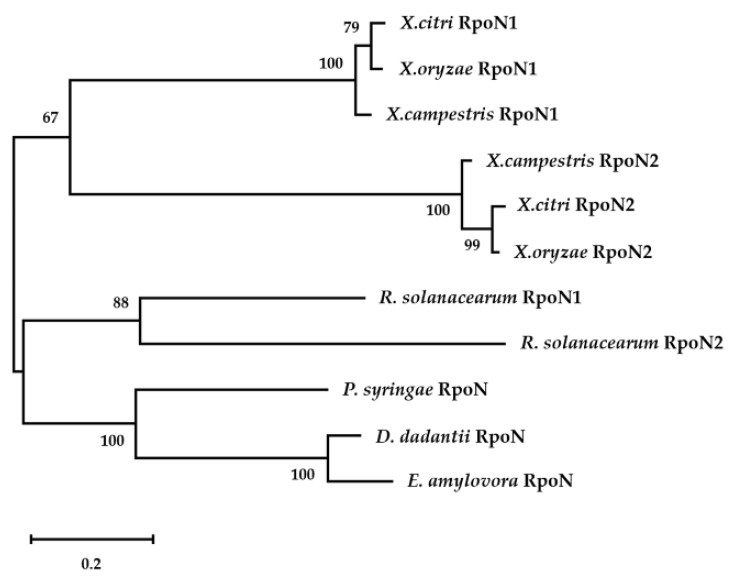
Phylogenetic analysis of RpoN. Eleven full-length RpoN protein sequences of *X. oryzae*, *X. citri*, *X. campestris*, *R. solanacearum*, *P. syringae*, *D. dadantii*, and *E. amylovora* were downloaded from the National Center for Biotechnology Information. A maximum likelihood (ML) tree was generated using MEGA-X with 1000 bootstrap values. Values on each branch represent the corresponding bootstrap probability. The scale bar indicates the number of amino-acid substitutions per site.

**Figure 3 ijms-22-12692-f003:**
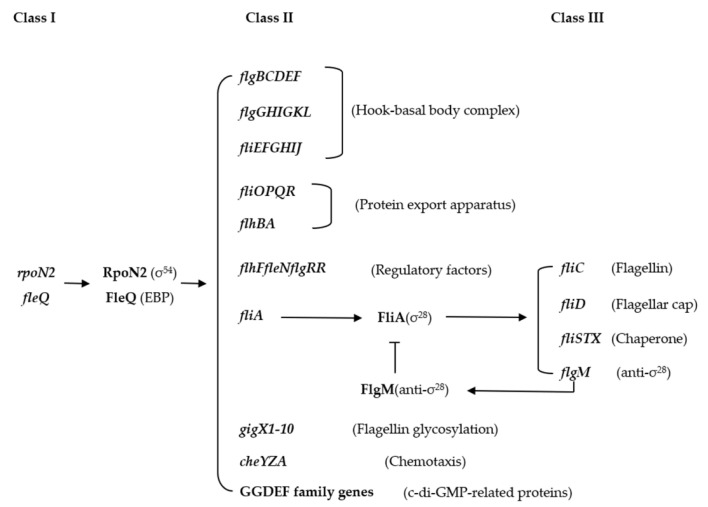
The three-tiered hierarchy of flagella synthesis in *Xoo*. The RpoN2 and FleQ, encoded by the class I genes *rpoN2* and *fleQ*, are the master regulators and control the transcription of class II genes. The class II gene products include most of flagellum structural components, regulatory factors (FlhF, FleN and FlgRR), flagellin glycosylation-related proteins (GigX1-10), chemotaxis-related proteins (CheYZA), c-di-GMP synthesis and degradation related proteins, and alternative sigma factor FliA. FliA regulates the transcription of class III genes, which encode the flagellin FliC, the flagellar cap FliD, the flagellar chaperone proteins FliS and FliTX, and the anti-σ^28^ factor FlgM. Interestingly, FlgM negatively regulates FliA activity and protects it from degradation.

**Figure 4 ijms-22-12692-f004:**
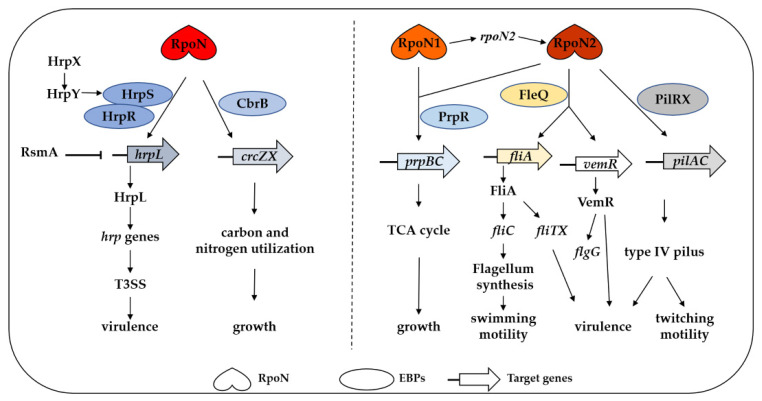
The regulatory functions of RpoN in major phytopathogenic bacteria. Left, *P. syringae* containing a single RpoN. RpoN requires HrpS and HrpR to activate HrpL-dependent transcription of T3SS, thereby regulating virulence. The two-component system HrpXY and RsmA is involved in the T3SS gene expression by positively regulating *hrpS* and negatively regulating *hrpL*. It also regulates *crcZX* by binding with CbrB and regulates bacterial growth by affecting carbon and nitrogen utilization. Right, *Xanthomonas* containing two copies of RpoN factors, RpoN1 and RpoN2, with unique and overlapping regulatory roles. RpoN2 regulates the synthesis of flagellum and T4P by interacting with FleQ and PilRX, respectively. It also depends on FleQ to regulate the *fliTX* and *vemR* expression and positively regulate bacterial virulence. Both RpoN1 and RpoN2 are involved in the regulation of bacterial growth by interacting with PrpR. In addition, RpoN1, along with unknown EBP, indirectly regulates the RpoN2 transcription.

**Table 1 ijms-22-12692-t001:** The RpoNs and EBPs in major phytopathogenic bacteria.

Bacteria	σ^54^ Factors	EBPs	Target Genes	Functions
*Xanthomonas oryzae*	RpoN1	PrpR	*prpBC*	Growth, virulence [59]
	RpoN2	FleQ, PilRX, PrpR	*fliA*, *fliC*, *fliTX*, *pilA*, *pilC*, *prpBC*	Growth, swimming, twitching, virulence [61,62,70]
*Xanthomonas campestris*	RpoN1	-	-	DSF, branched-chain fatty-acid production [58]
	RpoN2	FleQ	*fliA*, *fliC*	Swimming, flagellum synthesis [57]; biofilm, EPS, virulence [58]
*Xanthomonas citri*	RpoN1	-	-	Swimming, virulence, growth [60]
	RpoN2	FleQ	*fliC*;*flgG*	Swimming, virulence, growth [60,71]
*Ralstonia solanacearum*	RpoN1	PehR	*pilA*	Twitching, growth, virulence [63,64]
	RpoN2	-	-	-
*Pseudomonas syringae*	RpoN	HrpR, HrpS, CbrB	*hrpL*,*crcX*, *crcZ*	T3SS [65,72];virulence [66]; coronatine biosynthesis [73]; carbon source utilization, growth [74]
*Erwinia amylovora*	RpoN	HrpS	*hrpL*, *ihfA*, *rsmB*	T3SS, motility, growth, virulence [67,68,75,76]
*Dickeya dadantii*	RpoN	HrpS	*hrpL*	T3SS, virulence [69]

Note: “-” represents not found.

## Data Availability

Not applicable.

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
