# Peer review of "The Regulatory Functions of σ54 Factor in Phytopathogenic Bacteria"

_ijms, 2021, doi:10.3390/ijms222312692_

Round 1

Reviewer 1 Report

The manuscript by Yu et al., is a nice and comprehensive review about sigma54 and EBPs in phytopathogenic bacteria, with special consideration for RpoN functions in bacterial virulence.

Specific points:

1) Introduction, 1st paragraph. First, sigma classes are described as sigma70 and sigma54 families (which is correct), but then (lines 39-40) the authors state that generally sigma70 factors participate in transcription of only the housekeeping genes (or so that is how the text reads). This is oversimplification. Within the sigma70 family, there are many alternative sigma factors (such as sigmaS, E, F, H, FecI, in E. coli) that transcribe genes induces under different stress conditions. Please revise.

2) line 101 - it says that "ATP hydrolysis opens the RpoN-RNAP complex and initiates...". This is oversimplification again. The complex that is opened is the one formed between RpoN-RNAP and DNA, not just sigmaN and core RNAP.

3) The manuscript requires careful English language editing.

Author Response

1) Introduction, 1st paragraph. First, sigma classes are described as sigma70 and sigma54 families (which is correct), but then (lines 39-40) the authors state that generally sigma70 factors participate in transcription of only the housekeeping genes (or so that is how the text reads). This is oversimplification. Within the sigma70 family, there are many alternative sigma factors (such as sigmaS, E, F, H, FecI, in E. coli) that transcribe genes induces under different stress conditions. Please revise.

Response: We revised the sentence according to reviewer’s suggestion. Please see Lines 41-45: “In general, σ70 factors regulate the transcription of target genes by recognizing the -35/-10 promoter site (upstream from the transcription start site), while σ54 factors regulate the transcription of target genes by recognizing the highly conserved sequence GGN10GC at −24/−12 promoter site.”

2) line 101 - it says that "ATP hydrolysis opens the RpoN-RNAP complex and initiates...". This is oversimplification again. The complex that is opened is the one formed between RpoN-RNAP and DNA, not just sigmaN and core RNAP.

Response: Thank you for pointing this out. We correct it to RpoN-RNAP-DNA complex. Please see Line 119: “ATP hydrolysis opens the RpoN-RNAP-DNA complex and initiates RNA synthesis.”

3) The manuscript requires careful English language editing.

Response: We have edited the language thoroughly under the help of a professional English editing service. Please see the revised version.

Reviewer 2 Report

I found this review "The regulatory functions of σ54 factor in phytopathogenic bacteria" to be of interest to the readers of IJMS journal. 

Author Response

I found this review "The regulatory functions of σ54 factor in phytopathogenic bacteria" to be of interest to the readers of IJMS journal. 

Response: We appreciate for your approval.

Reviewer 3 Report

The review is dedicated to the functions of sigma54 factor. It is an important transcriptional regulator, which is present in genomes of many bacteria and regulates several hundred genes. Though it seems a very interesting topic for a review, the manuscript has several flaws. The manuscript contains a lot of details, but is written in an awkward way, which makes it very difficult to follow. The facts are provided practically without any system, with many gene names, as an enumeration, but without explanation, what and why is important and how the facts are connected with each other. The only try to systemize the data is provided in the Conclusion section, but what is called a simple model, is not that in fact. It is just a list of functions, regulated by one or two sigma factors. Though the review is dedicated to the phytopathogenic bacteria, there is no explanation, why they are important, or no comparison of phytopathogenic and other bacteria are provided.

Author Response

The review is dedicated to the functions of sigma54 factor. It is an important transcriptional regulator, which is present in genomes of many bacteria and regulates several hundred genes. Though it seems a very interesting topic for a review, the manuscript has several flaws. The manuscript contains a lot of details, but is written in an awkward way, which makes it very difficult to follow. The facts are provided practically without any system, with many gene names, as an enumeration, but without explanation, what and why is important and how the facts are connected with each other. The only try to systemize the data is provided in the Conclusion section, but what is called a simple model, is not that in fact. It is just a list of functions, regulated by one or two sigma factors. Though the review is dedicated to the phytopathogenic bacteria, there is no explanation, why they are important, or no comparison of phytopathogenic and other bacteria are provided.

Response: Thanks for your suggestion. We further improved the manuscript to make it more readable and more accurate.

  1. About the organization and the structure of the manuscript. σ54 factor regulates the transcription of target genes in dependent on enhancer-binding proteins (EBPs). However, many documents reported the biological function of σ54 in phytopathogenic bacteria but little about the intrinsic regulatory mechanism of σ54 and EBPs, and the reciprocal regulation between σ54 factors and their interaction with EBPs are also largely unknown in double σ54 factors bacteria. Therefore, it is not easy to systematically elucidate the regulatory network of σ54 factors in phytopathogenic bacteria. In this review, in order to better introduce the research progress of σ54 factors in phytopathogenic bacteria, we firstly reviewed σ54 factors and main EBPs in important phytopathogenic bacteria. Secondly, we introduced detailly the regulatory function of σ54 factors and EBPs in motility, growth and virulence in order. Finally, we summarized the regulatory network of single or double σ54 factors and put forward the future works and direction in Conclusions and Future Perspectives.
  2. About the model. We replaced the original sentence “we propose two simple regulatory models of RpoN” with “we schematize the regulatory mechanism of RpoN in P. syringae and Xanthomonas species, which have single and double σ54 factors, respectively” in the Conclusion section (Lines 375-376).
  3. About the large number of EBPs and gene names. To make it easier for readers, we added simple annotations when they first appeared, such as
    Lines 127-128: “……FleQ, an important EBP, and regulates flagellar-associated gene transcription”;

Lines 129-130: “PilRX, an EBP type, is located in the T4P gene cluster and forms a two-component system with PilSX”;

Lines 133-134: “PrpR is a propionate catabolism operon regulatory protein”;

Line 136: “PXO_03020, PXO_03564 and PXO_03965 are NtrC family proteins”;

Lines 142-143: “which depends on two T3SS regulators, HrpS and HrpR, to regulate……”;

Lines 190-196: “……and σ28 factor FliA ……flagellin protein FliC, flagellar cap protein FliD, flagellar chaperone proteins FliS and FliTX, and anti-σ28 factor FlgM…...flagellin glycosylation-related genes (gigx1-gigx10), chemotaxis-associated genes (cheYZA) and c-di-GMP-related genes (PXO 06199, PXO 06201, and PXO 06202)”;

Lines 232-233: “CbrB, which is an EBP of RpoN and belongs to the NtrC family of response regulators”;

Lines 233-234: “CrbB binds to CrbA and forms a two-component system”;

Lines 239-240: “CbrAB directly activates the transcription of sRNAs (crcZ and crcY)”;

Line 242: “Hfq-Crc, the key regulator of the carbon catabolite repression (CCR) process”;

Lines 268-269: “prpB and prpC, which encode methylisocitrate lyase and 2-methylcitrate synthase, respectively”;

Lines 278-279: “The alternative sigma factor, HrpL, is the primary transcription factor that controls the expression of T3SS-associated genes”;

Line 306: “……integration host factor (IHF), a nucleoid-associated protein……”;

Lines 324-325: “including the major pilin gene, pilAX, and the inner membrane platform protein gene, pilCX”;

Lines 332-333: “……PehR, which is one of EBPs and forms a two-component system with PehS”;

Lines 336-337: “FlhDC, the primary regulator of flagellum synthesis”.

In addition, to emphasize the important role of flagella, we added a sentence “Flagella are sophisticated organelles found in many bacteria where they perform functions related to motility, signal detection, biofilm formation, colonization and attachment to host tissues.” Please see Lines 179-181.

  1. About the importance of the phytopathogenic bacteria. We added a paragraph to describe the importance of the selected phytopathogenic bacteria. We chose these bacteria because they are the most important pathogenic bacteria in crops. At the same time, there are also some reports on the regulatory roles of sigma 54 in these pathogenic bacteria. Please see

Lines 87-104: “Phytopathogenic bacteria, like fungi and viruses, cause economically important plant diseases and serious threat to world food security. P. syringae, Ralstonia solanacearum, Xanthomonas species, Erwinia amylovora and Dickeya dadantii are the most important phytopathogenic bacteria. P. syringae causes important crop diseases, and it is a well-known model organism for plant-pathogen interaction-related study. R. solanacearum is probably the most destructive pathogen worldwide, and it has a very broad host range that can infect 200 plant species belonging to over 50 plant families. X. oryzae pv. oryzae (Xoo) and oryzicola (Xoc) are the most important bacterial pathogens of rice, resulting in a 20%-50% loss of crop yield. X. campestris pv. campestris (Xcc) is the causative agent of black rot of crucifers and affects cultivated brassicas worldwide. E. amylovora causes fire blight disease of apple, pear, quince, blackberry and raspberry, and threatens the safe production of the major fruits. D. dadantii causes disease mainly in tropical and subtropical environments and has a wide host range, including Saintpaulia and potato. In this review, we summarized the recent research on the σ54 factor and their regulatory functions in these phytopathogenic bacteria to enhance the current understanding of the regulatory mechanism of phytopathogenic bacteria's motility, growth, and virulence.”